# *In Vitro* High-Throughput Genotoxicity Testing Using γH2AX Biomarker, Microscopy and Reproducible Automatic Image Analysis in ImageJ—A Pilot Study with Valinomycin

**DOI:** 10.3390/toxins15040263

**Published:** 2023-04-01

**Authors:** Bára Křížkovská, Martin Schätz, Jan Lipov, Jitka Viktorová, Eva Jablonská

**Affiliations:** 1Department of Biochemistry and Microbiology, University of Chemistry and Technology, Prague, Technická 5, 166 28 Prague 6, Czech Republic; 2Department of Mathematics, Informatics and Cybernetics, University of Chemistry and Technology, Prague, Technická 5, 166 28 Prague 6, Czech Republic

**Keywords:** genotoxicity, *in vitro* testing, high-throughput, bioimage analysis, ImageJ

## Abstract

(1) Background: The detection of DNA double-strand breaks *in vitro* using the phosphorylated histone biomarker (γH2AX) is an increasingly popular method of measuring *in vitro* genotoxicity, as it is sensitive, specific and suitable for high-throughput analysis. The γH2AX response is either detected by flow cytometry or microscopy, the latter being more accessible. However, authors sparsely publish details, data, and workflows from overall fluorescence intensity quantification, which hinders the reproducibility. (2) Methods: We used valinomycin as a model genotoxin, two cell lines (HeLa and CHO-K1) and a commercial kit for γH2AX immunofluorescence detection. Bioimage analysis was performed using the open-source software ImageJ. Mean fluorescent values were measured using segmented nuclei from the DAPI channel and the results were expressed as the area-scaled relative fold change in γH2AX fluorescence over the control. Cytotoxicity is expressed as the relative area of the nuclei. We present the workflows, data, and scripts on GitHub. (3) Results: The outputs obtained by an introduced method are in accordance with expected results, i.e., valinomycin was genotoxic and cytotoxic to both cell lines used after 24 h of incubation. (4) Conclusions: The overall fluorescence intensity of γH2AX obtained from bioimage analysis appears to be a promising alternative to flow cytometry. Workflow, data, and script sharing are crucial for further improvement of the bioimage analysis methods.

## 1. Introduction

The purpose of genotoxicity testing is to identify chemicals that can cause genetic alterations in cells [1]. In the last 50 years, a variety of *in vitro* and in vivo methods have been developed for the routine testing of chemicals, including genotoxicity testing. They comprise standardised procedures that are described in national and international guidelines (recently reviewed in [2,3]). There is substantial effort to fulfil the 3R concept (replacement, reduction, and refinement), i.e., to reduce in vivo experiments and to use human-relevant and reliable *in vitro* methods [4]. However, currently standardised *in vitro* methods for genotoxicity testing using mammalian cells (HPRT and TK assays, micronucleus assay) are tedious and not suitable for high-throughput testing. One of the promising markers of DNA damage response for genotoxicity assessment seems to be the phosphorylation of the histone variant H2AX [5].

H2AX represents the constitutively expressed and evolutionarily conserved variant of the H2A protein family that is a component of the histone octamer in nucleosomes. The phosphorylation event at serine 139 in mammals, referred to as γH2AX, is connected to double stranded breaks (DSBs) and is one of the first reactions. This will trigger a chromatin condensation that appears to play a crucial role in the recruitment of damage signalling or repair factors to the site of the damage. The γH2AX could be the response not only to DSBs, but also to single-strand breaks, during replication stress, or to the fragmentation during cell apoptosis [6,7,8].

*In vitro* measurement of the γH2AX is an increasingly popular method of measuring genotoxicity in combination with other classical methods. The main advantages are the specificity and sensitivity of the system [3,5,9] and the list of immunoassays suitable for the γH2AX detection: Western blot, ELISA, flow cytometry, and immunofluorescence microscopy, the last mentioned having the highest sensitivity [3,10].

Antibodies or commercial kits for γH2AX detection are available and suitable for flow cytometry as well as for microscopic analysis and other novel immuno-based methods [11]. Flow cytometry only allows us to measure overall fluorescence, whereas γH2AX detection by immunofluorescence microscopy is based either on the foci detection per cell (using a confocal microscope) or on the quantification of the overall intensity from images of lower magnification [10], the second mentioned approach being more suitable for high-throughput analysis in a 96-well plate format. With immunofluorescence, high-content screening (multiparametric analysis) is also possible [12]. Nuclei are routinely counterstained (by Hoechst 33342 or DAPI) and, in addition, cell viability (i.e., membrane integrity) can be simultaneously monitored using appropriate DNA-binding dyes (e.g., Image-iT DEAD Green™ or 7-AAD) or combined with other biomarkers such as 53BP1 (p53 binding protein 1) [13]. Nevertheless, the multiparametric analysis increases the total time of analysis.

Taking into account that the wide-field microscope is more commonly a part of biological laboratory equipment compared to the flow cytometer, bioimage analysis of the overall γH2AX signal appears to be an assay of choice for high-throughput genotoxicity screening of substances. In addition, fixed plates can be stored in a freezer for a short time and re-analysed or the original images can be databased and reanalysed, if necessary.

Various authors used γH2AX detection by immunofluorescence microscopy; however, they either perform an automatic analysis on a specialised device with dedicated software or they use a commercial software (Table 1). Studies using open-source software for bioimage analysis are sparse and a detailed description of semiautomatic or automatic image analysis workflow is usually not provided.

Open-source bioimage analysis tools are highly preferred, as they enable reproducibility. Commercial platforms often focus on ease-of-use, but the details of the image processing algorithms are hidden. On the contrary, the details are completely transparent in open-source platforms [20]. Open source bioimage informatics tools (CellProfiler, Icy, ImageJ) for the analysis of DNA damage and associated biomarkers were reviewed in [21]. ImageJ is often a tool of choice, because of its long existence, wide adoption, and extensible plugin architecture, which means that anyone can access and modify the source code and everybody has access to a wealth of resources, such as tutorials and plugins. It also allows users to save the analysis steps as scripts, which can be rerun to obtain the same results on different data, ensuring that the results are reproducible [20].

In this study, we present a detection of DSBs in adherent mammalian cells in 96-well format using a commercially available kit (HSC Damage Kit, Invitrogen, Waltham, MA, USA [22]) for better reproducibility, and the γH2AX response is evaluated using an inverted wide-field fluorescence microscope. Valinomycin is used as a model genotoxic substance causing γH2AX positive response and is recommended by the kit manufacturer [22]. Image acquisition is automated (autofocus, motorised stage). The viability is evaluated from the area of Hoechst 33342-stained nuclei (non-viable cells are washed during the procedure and are not present) and the signal from γH2AX is measured as immunofluorescence intensity in the Cy3 channel. The last component of the kit, Image-iT DEAD Green, is not used in order to accelerate the image acquisition and to minimise the potential risk of leakage of the signal to other channels. The image automatic analysis is performed in the open-source software ImageJ.

## 2. Results

We obtained microscopic images of immuno-stained CHO-K1 and HeLa cells after incubation (two time points) with valinomycin in two concentrations. The mean fluorescence values, MFV (γH2AX signal from Cy3 channel), were extracted using the segmented nuclei from the DAPI channel and expressed as area-scaled relative fold change in fluorescence over the control (Equations (1) and (2)), alongside cytotoxicity changes (% control), expressed as Relative Area (RA, Equation (3)). These data were used to evaluate whether a compound induced a positive, negative, or equivocal response in the assay (scaling was adopted from Smart et al. [23], Table 2). The results are summarised in Table 3 for the wells after 4 h and in Table 4 for the wells after 24 h. Representative images are shown in Figure 1 and Figure 2. Boxplots are shown in Appendix A.

The fluorescence microscopy analysis validated a time-dependent increase in Cy3 nuclei brightness after treatment, which was accompanied by a decrease in the relative area of cells based on the measured area of segmented nuclei. These changes were consistent with an expected genotoxic and cytotoxic effect of the treatment on the cells, respectively.

All RA values (analogue of relative cell counts) were above the limit of 25% which means that genotoxicity can be evaluated. Fold changes after 4 h treatment were below 1.5 in the case of HeLa cells and were above 1.5 in the case of CHO-K1 cells. Fold changes after 24 h treatment were above 1.5 in all cases, which indicates genotoxicity (Table 2 and Table 3).

## 3. Discussion

Here, we propose a method for high-throughput genotoxicity testing based on γH2AX detection using immunofluorescence microscopy and automated bioimage analysis in open-source software.

In this pilot study, we made several simplifications:We are aware of the fact that for the measurement of γH2AX response, cell lines derived from normal tissues are considered to be more reliable than those from cancer tissues [24]. Nevertheless, HeLa cell line was used as a model since it is recommended in the manual of the kit’s manufacturer. The CHO-K1 cell line was used as the second cell type because it is a cell line used for the HPRT mutation assay accepted by OECD and EPA.As a model compound, we only used valinomycin because it was used by the manufacturer of the kit. There was, therefore, no need to use metabolic activation, because valinomycin is genotoxic per se.To accelerate image acquisition and to minimize the potential risk of channel-to-channel leakage, we presumed that dead cells were washed away during the preparation of the samples and, therefore, we did not use any additional dye to detect dead cells.We used nuclei area, not number of nuclei, to evaluate cell mass.

The general limitations are listed in Appendix B.

The idea was to offer an alternative to γH2AX detection by flow cytometry [23]. Using this method, Smart et al. measured median FL1 fluorescence values (γH2AX) and evaluated the fold change in γH2AX signal and relative cell counts. Inspired by [25], we used similar parameters, i.e., mean fluorescence intensity values, MFV (γH2AX signal from Cy3 channel) in a region of interest (ROI), and area-scaled relative fold change in γH2AX fluorescence over the negative control. Our workflow was similar to that of [9]; however, contrary to that study, we provide a detailed description in an open-source software ImageJ.

Instead of relative cell counts, we expressed cytotoxicity as relative area (RA) of the nuclei. As already mentioned, we did not directly quantify the number of nuclei in each well (mainly due to problems with nuclei on the edges); nevertheless, we propose that nuclei area is an easier and sufficient alternative to evaluate a cell mass in a well.

We tried to compare our values with those from the commercial platform listed in the manufacturer’s protocol [22] of the commercial kit we used. The same model compound (30µM valinomycin) and incubation time (24 h) were used; however, the cell lines were different. We digitalised the graph from the protocol (values in Table 5). γH2AX response is expressed as mean average intensity (γH2AX nuclear intensity); the analogous quantity MFV presented by us. The mean intensity over cell numbers, fold, and relative cell numbers were calculated similarly as MoA, fold, and RA (Equations (1)–(3)). These results (cytotoxicity as well as the genotoxicity) are comparable with our results from CHO-K1 cell line (Table 4).

We observed differences between the two cell lines used. Valinomycin was more cytotoxic to the CHO-K1 cell line compared to the HeLa cell line. It is in agreement with the studies confirming the rodent cell lines being consistently more susceptible to cytotoxicity [26]. The fold change after 4h incubation was higher in CHO-K1 cells probably because they rank among p53-compromised cell types [26]. Despite the recommendations of the kit manufacturer, the HepG2 cell line is often used, which not only offers the potential for metabolic activation of pro-mutagens (production of liver enzymes), but also meets the requirements for human non-tumour tissue lines [27]. The results obtained in this way can better correspond to real conditions [28,29].

Compared to flow cytometry, we observed higher fold changes after 24 h, especially in HeLa cells. The high score after 24 h can be false positive due to increased apoptosis, because DNA degradation as a consequence of apoptosis is known to induce γH2AX [15,30], especially after incubation with valinomycin [31]. Moreover, the high toxicity of valinomycin and its mechanism of action suggest that it is not a suitable positive control for such genotoxicity measurements [32].

The number of cells evaluated was lower in our case of microscopic analysis (1000–3000 objects per well) compared to flow cytometry (10^4^ nuclei [23]). This probably caused the higher deviations (or higher interquartile change in our case).

Overall, our results suggest that the microscopy-based validation of genotoxicity on the cells is consistent with other methods [9,23]. The detection of overall fluorescence intensity using microscopic bioimage analysis seems to be more sensitive than by using a spectrophotometer, as in [33]. In this case, results may be inaccurate due to non-specific antibody binding, or, in the case of foci, the signal may be so weak that it can easily be confused with noise.

## 4. Conclusions

Overall fluorescence intensity of γH2AX obtained from bioimage analysis appears to be a promising alternative to flow cytometry in genotoxicity testing *in vitro*. Workflow, data, and script sharing are crucial for further improvement of the bioimage analysis methods. Our future work will be based on the implementation of more cell models, recruitment of metabolic activation by S9 mix and more model compounds, and finally on the validation of this workflow within laboratories.

## 5. Materials and Methods

### 5.1. Chemicals

Bovine serum albumin (BSA A7906; Merck, Germany); Dimethylsulfoxide (DMSO D8418; Merck, Germany); Dulbecco’s Modified Eagle’s Medium—high glucose (DMEM D0819; Merck, Germany); Foetal bovine serum (FBS F7524; Merck, Germany); HCS DNA Damage Kit (H10292; Invitrogen, USA); L-Proline (P5607; Merck, Germany); Minimum Essential Medium (MEM M0446; Merck, Germany); MEM Non-essential Amino Acid Solution 100x (NEAA M7145; Merck, Germany); Paraformaldehyde 16% (043368.9M; Thermo Scientific Chemicals, UK); Valinomycin (V0627; Merck, Germany).

### 5.2. Cell Lines and Culture Conditions

Human cervical adenocarcinoma cell line (HeLa, CCL-2) was obtained from American Type Culture Collection (ATCC, Manassas, VA, USA), and Chinese hamster ovary cell line (CHO-K1 85051005) was obtained from the European Collection of Authenticated Cell Cultures (ECACC, Porton Down, UK). The HeLa cells were grown in MEM supplemented with 10% FBS and NEAA. CHO-K1 cells were cultivated with DMEM supplemented with L-proline (final concentration 35 mg/L). The cell incubation took place in a humidified atmosphere of 5% CO_2_ at 37 °C.

### 5.3. Direct Measurement of DNA DSBs

The cells were seeded in a concentration of 0.5 × 10^5^ cells/mL in the 96-well plate (VWR, 10062-900). The cells were rinsed by phosphate buffered saline (PBS) after 24 h incubation, and medium with reduced FBS content (5%) was added. Valinomycin was dissolved in DMSO and added to cells in two final concentrations (30 and 15 µM). Final concentration of DMSO in medium was 1%. Medium with 1% DMSO was used as a negative control. After 4 h/24 h incubation, the visualization was performed using the following protocol of the HCS DNA Damage Kit, Invitrogen, Waltham, MA, USA [22]. The cells were fixed by 4% paraformaldehyde solution for 15 min at room temperature. The cells were rinsed once by PBS and the permeabilization was performed using Triton X-100 solution, by incubation for 15 min at room temperature. The wells were rinsed with PBS once and the plate was blocked by 1% BSA blocking solution. After 1 h incubation at room temperature, the blocking solution was removed and 100 µL of pH2AX mouse monoclonal antibody solution (1:1000 in 1% BSA) was pipetted into each well and incubated for 1 h at room temperature. After three rinses by PBS, the 100 µL of Alexa Fluor 555 goat anti-mouse IgG (H+L; 1:2000) and Hoechst 33342 (1:6000) solution was added and the plate was incubated for 1 h at room temperature and protected from light. After the incubation, the wells were rinsed three times by PBS. The plate was stored with 100 µL of PBS in the refrigerator (4 °C) until the image analysis was performed.

### 5.4. Measurement Settings

Manufacturer and model of microscope: Olympus IX83 P2ZF;Objective lens magnification: 10×; NA = 0.3;Excitation filters (mounted in the light source);Violet: 395/25 nm; LED module 1, DAPI;Green: 555/28 nm; LED module 5, Cy3;Quad band filter set for DAPI/FITC/Cy3/Cy5;Quad band polychroic mirror (mounted in the filter turret);BP 411–454 nm;BP 495–536 nm;BP 577–617 nm;BP 655–810 nm;Emission filters (mounted in the fast emission filter wheel, in front of the camera);DAPI: BP 421–445 nm;Cy3: BP 581–619 nm;Illumination light source: Lumencor Spectra X Lamp;Camera manufacturer and model: Hamamatsu ORCA-Flash4.0;Pixel size: 650 nm × 650 nm;Software program(s) and version: OLYMPUS cellSens Dimension 3.2 (Build 23706);Image acquisition settings including exposure times, gain, and binning: exp 500 ms, gain: 0, binning: 4 × 4;Experiment manager: ZDC + autofocus, two channels: DAPI and Cy3 (Figure 3);Well navigator: single frames, 4 × 4 per well (Figure 4).

### 5.5. BioImage Analysis

A reproducible image analysis workflow is a collection of tools and resources that enable anyone to replicate the process of handling and analysing images, with the goal of obtaining the same results. This workflow package consists of three key components:A description of the workflow;The code of the workflow;The original image data.

#### 5.5.1. Description of the Workflow

In this paper, we present a reproducible image analysis workflow [34,35] to allow replication of the image handling and analysis process. The workflow is based on the idea of Smart et al. [23]. The authors used flow cytometry and introduced useful genotoxicity evaluation criteria for the γH2AX (Table 2). “Median FL1 (H2AX) fluorescence values were extracted and expressed as relative fold change in H2AX fluorescence over control, alongside cytotoxicity RCC changes (% control); these data were used to determine whether a compound induced a positive, negative or equivocal response in the assay.” [23].

Our aim was to adopt these evaluation criteria in connection with automated immunofluorescence microscopy. We focus on the thresholding of DAPI stained nuclei in channel 2 and brightness value analysis in the Cy3 channel. The main parameters of interest for analysis are the area of each nuclei object and the mean brightness of each object from the Cy3 channel, which are used for comparison across wells.

The workflow consists of three main steps: sorting of data, image analysis, and feature analysis. The sorting step uses an ImageJ macro script that expects images to be named following the convention “ChannelName_YYYYMMD_Well_PositionInWell_AcqRun.FileFormat” and creates a set of subfolders for each well. The image analysis step is performed using another ImageJ macro, which detects nuclei through automatic Otsu thresholding and computes the area and brightness for each object. Finally, the feature analysis step is performed in Python using a Jupyter Notebook, where specified wells are statistically compared, and fold changes are calculated.

#### 5.5.2. The Code of the Workflow and the Original Image Data

The complete workflow package, including all scripts and notebooks, is publicly available through published data and GitHub, allowing for the easy replication and verification of results.


**ImageJ macro-overview**


This protocol documents an image data flow utilised and inspired by CLIJx-Assistant [36].

We start our image data flow with image_1, a DAPI channel with nuclei.Following this, we apply “Copy” on image_1 and obtain a new image out, image_2.As the next step, we apply “Otsu” auto threshold on image_2 and obtain a new image mask out, image_3. The threshold values are saved and used on all DAPI images from the same well.Afterward, we apply “Analyze Particles” on image_3, and single out a region of nuclei as a Region of Interest (ROI) set. All ROIs touching edges are skipped.In the next step, we open image_4, which is the Cy3 channel. We apply “Copy” on image_4, and obtain a new image, image_5.We apply background subtraction with s rolling ball of size 50 on image_5 to subtract the local background value from intensity measurements, and obtain image_6.Afterward, image_6 is selected for measuring features under ROIs from the previous step.The process Log and measured features from Cy3 channel for the whole well and summary are saved in the “Results” subfolder in the CSV table. A flattened image_4 with ROIs outlined is saved as a JPEG for later inspection.

The macro logs version of ImageJ and BioImage plugin version on each run was tested in ImageJ version 1.53t99 [37]. The logs also contain information about image size, count of objects, and threshold values.

While using the ImageJ Macro Markdown plugin, it is possible to extract a protocol documenting the analysis of the selected well. The code for the version used in this paper is available on Zenodo and on GitHub (links in a section “Data and software availability statement”).

### 5.6. Calculations

MoA (Mean fluorescence value over Area) is calculated as
(1)$$\mathrm{MoA}=\frac{\mathrm{MFV}}{\mathrm{area}}$$

Fold is calculated as
(2)$$\mathrm{fold}=\frac{{\mathrm{MoA}}_{\mathrm{sample}}}{{\mathrm{MoA}}_{\mathrm{ctrl}}}$$

RA (relative area) is calculated as
(3)$$\mathrm{RA}=\frac{{\mathrm{Area}}_{\mathrm{sample}}}{{\mathrm{Area}}_{\mathrm{ctrl}}}\times 100$$

### 5.7. Data Analysis and Statistics

The measured features obtained from BioImage Analysis were examined for normal distribution, and the findings are shown in Appendix A. The results indicate that none of the measured data follow a normal distribution; non-parametric measures were therefore employed. Mainly, the interquartile range (IQR) was utilised as a measure of dispersion instead of the standard deviation (STD) typically used. As for the relative area and folds (calculated as the brightness values of the treated sample over the control values), they were presented without IQR since only a single statistically descriptive value was used.

## Figures and Tables

**Figure 1 toxins-15-00263-f001:**
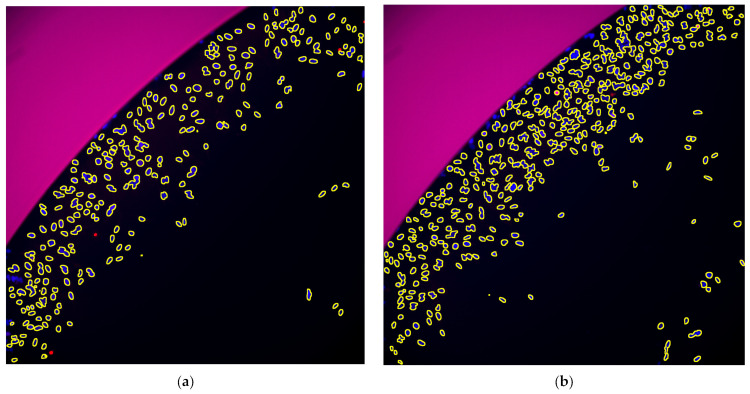
Processed images after **4 h**. Red: Cy3 (signal from γH2AX), Blue: DAPI (nuclei), Yellow outlines: detected objects. Images were enhanced using ImageJ run (“Enhance Contrast”, “saturated = 0.35”). Same contrast transform was applied on control and treatment, channel wise. (**a**) HeLa, val15, well B8; (**b**) HeLa ctrl, well B6. (**c**) CHO-K1, val15, well B4; (**d**) CHO-K1 ctrl, well B2.

**Figure 2 toxins-15-00263-f002:**
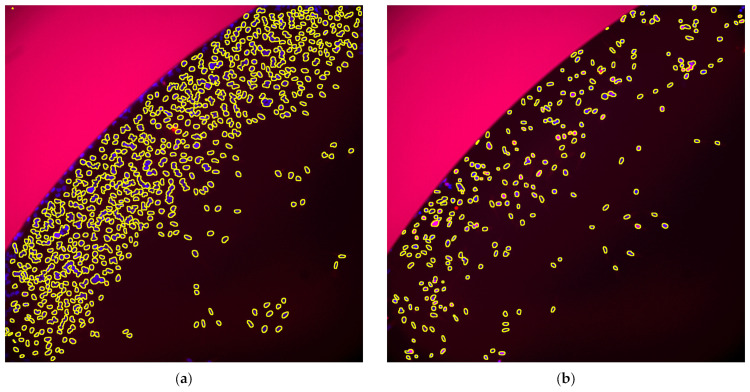
Processed images after **24 h**. Red: Cy3 (signal from γH2AX), Blue: DAPI (nuclei), Yellow outlines: detected objects. Images were enhanced using ImageJ run (“Enhance Contrast”, “saturated = 0.35”). Same contrast transform was applied on control and treatment, channel wise. (**a**) HeLa, val15, well C8; (**b**) HeLa, ctrl, well C6. (**c**) CHO-K1, val15, well C4; (**d**) CHO-K1 ctrl, well C2.

**Figure 3 toxins-15-00263-f003:**
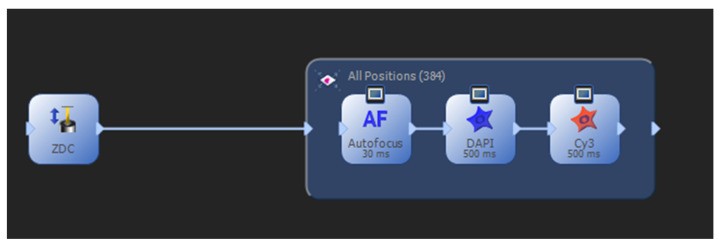
Scheme of experiment manager.

**Figure 4 toxins-15-00263-f004:**
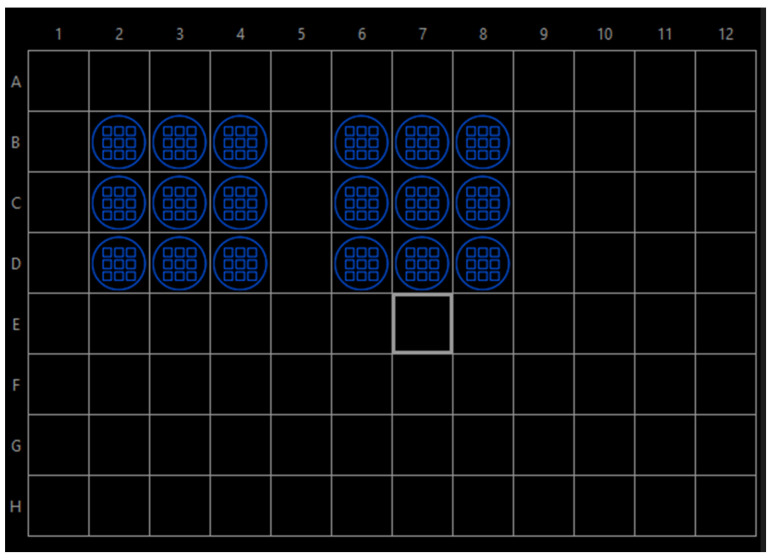
Scheme of well navigator in OLYMPUS cellSens Dimension software. Blue marks indicate the layout of the samples in a 96-well plate.

**Table 1 toxins-15-00263-t001:** Example of studies using immunomicroscopic γH2AX detection for genotoxicity evaluation.

Overall Intensity (OI) or Foci Detection (FD)	Device for Detection	Software for Bioimage Analysis	Ref.
FD	AKLIDES Cell Damage system, Medipan		[14]
FD	AKLIDES Cell Damage system, Medipan		[10]
OI	Cellomics Arrayscan VTI platform ^1^, Thermo Scientific, TargetActivation Bioapplication software V.6.6.1.4.		[9]
OI	ArrayScan VTI HSC reader Reader ^1^, Thermo Scientific		[15]
OI	Cellomic Arrayscan VTI HCS Reader ^1^, Thermo Scientific		[16]
FD	confocal laser scanning microscope (Zeiss LSM 510)	Foci 8.0 software (Schultz and Belyaev, unpublished)	[17]
FD	confocal microscope (Nikon)	NE Element software (Nikon)	[18]
OI	Olympus fluorescence microscope (BX51)	ImageJ software (v.1.47)	[19]

^1^ updated version of the device is called CellInsight.

**Table 2 toxins-15-00263-t002:** Genotoxicity evaluation criteria for the γH2AX. Adopted from [23].

γH2AX Signal Increase	RCC ^1^	Classification
>1.5×	Above 25%	Genotoxic
<1.5×	0–100%	Non-genotoxic
>1.5×	Below 25%	Cytotoxicity-driven genotoxicity = false positive
1.5×	≥25%	Equivocal

^1^ RCC = relative cell counts.

**Table 3 toxins-15-00263-t003:** The values from bioimage analysis after **4 h** exposition of CHO-K1 and HeLa cell lines to valinomycin in two concentrations (30 and 15 µM) compared to unaffected control (ctrl). MFV = mean fluorescence values; IQR = interquartile range; MoA = Mean fluorescence values over Area of the nuclei; RA = Relative Area (analogue of RCC = relative cell count).

Cell Line	Sample	MFV (γH2AX Response)		Area (Nuclei) [µm^2^]		MoA	Fold	RA
		Mean	IQR	Mean	IQR			
CHO-K1	Ctrl	1.641	1.520	2.130	1.058	0.770	1	100%
	Val30	2.179	1.872	1.423	0.862	1.531	1.987	66.81%
	Val15	2.151	1.439	1.614	0.814	1.333	1.730	75.78%
HeLa	Ctrl	0.132	0.166	3.249	1.334	0.041	1	100%
	Val30	0.127	0.149	2.955	1.109	0.043	1.063	90.94%
	Val15	0.119	0.147	3.040	1.167	0.039	0.966	93.56%

**Table 4 toxins-15-00263-t004:** The values from bioimage analysis after **24 h** exposition of CHO-K1 and HeLa cell lines to valinomycin in two concentrations (30 and 15 µM) compared to unaffected control (ctrl). MFV = mean fluorescence values; IQR = interquartile range; MoA = Mean fluorescence values over Area of the nuclei; RA = Relative Area (analogue of RCC = relative cell count).

Cell Line	Sample	MFV (γH2AX Response)		Area (Nuclei) [µm^2^]		MoA	Fold	RA
		Mean	IQR	Mean	IQR			
CHO-K1	Ctrl	1.518	1.509	2.186	0.973	0.694	1	100%
	Val30	2.785	3.229	0.893	0.405	3.119	4.495	40.83%
	Val15	2.605	2.670	1.173	0.572	2.221	3.199	53.66%
HeLa	Ctrl	0.064	0.082	2.919	1.276	0.022	1	100%
	Val30	0.595	0.324	2.070	0.806	0.287	13.200	70.91%
	Val15	0.358	0.275	2.227	0.893	0.161	7.376	76.30%

**Table 5 toxins-15-00263-t005:** The values obtained from a chart from manufacturers protocol (evaluated using the Thermo Scientific Cellomics ArrayScan VTI and Compartmental Analysis Bioapplication) after 24 h exposition of A549 cell lines to 30 µM valinomycin compared to unaffected control (ctrl). γH2AX response is expressed as mean average intensity (γH2AX nuclear intensity)—the analogue of MFV; relative cell number is an analogue of RA = relative area.

Sample	γH2AX Response	Cell Number	Mean over Cell Number	Fold	Relative Cell Number
Ctrl	90.4	184	0.491	1.0	100%
Val30	201.5	72.9	2.764	5.6	40%

## Data Availability

All data are available on-line, see Appendix C.

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
