# Peer review of "In Vitro High-Throughput Genotoxicity Testing Using γH2AX Biomarker, Microscopy and Reproducible Automatic Image Analysis in ImageJ—A Pilot Study with Valinomycin"

_toxins, 2023, doi:10.3390/toxins15040263_

Round 1

Reviewer 1 Report

The novelty and the quality of the manuscript are good and it does not need extensive improvement before publication. It is carefully organized and written. It is easy to follow it and contains clear comments and conclusions.  In my opinion, this manuscript is very detailed and meticulous, it covers all the literature in the field with critical point of view. The topic have been completely covered and is well connected through the text. There is a significant  novelty in presented topic.  For all these reasons, I can recommend the acception of the manuscript after minor revision:

 1. I think that information about "In vitro measurement of the γH2AX" could be extended, more examples  should be added. This would be valuable for later publication citation.

 2. The superiority of  the immunofluorescence than flow cytometry and other methods should be more emphasized.

 3. The manuscript should be extended in scientific discussion. The authors presented their results and compared to some works, but did not present explanations for the reasons to reach these results.

 4. Not all of the described results are covered in the discussion section.

 5. No all information was given of automated bioimage analysis in open-source software.

Author Response

Dear sir or madame,

we have finished the revisions upon requests:

1) I think that information about "In vitro measurement of the γH2AX" could be extended, more examples  should be added. This would be valuable for later publication citation.

Answer

We have slightly extended the paragraph dealing with the measurement of the gamma-H2AX (row 52, row 62).

2) The superiority of  the immunofluorescence than flow cytometry and other methods should be more emphasized.

Answer

We have tried to point out also the advantages of fluorescence microscopy over flow cytometry in this type of experiment (row 67).

3) The manuscript should be extended in scientific discussion. The authors presented their results and compared to some works, but did not present explanations for the reasons to reach these results.

Answer

The core of the manuscript remains the set-up of automated bioimage analysis. Nevertheless, we have modified the discussion, trying to explain in more details our opinion about the meanings of the results obtained.

4) Not all of the described results are covered in the discussion section.

Answer

We have also tried to cover all the results presented in the discussion, mainly the differences in cell line used (row 187).

5) No all information was given of automated bioimage analysis in open-source software.

Answer

Regarding this comment: We have provided the detailed process of image analysis and it is available at Zenodo and GitHub. We are not sure, whether the link had to be removed during anonymization of the publication before sending to the reviewers. The final version with the link included is covering the details in full.

Reviewer 2 Report

In the paper entitled "In vitro high-throughput genotoxicity testing using γH2AX biomarker, microscopy and reproducible automatic image analysis in ImageJ - a pilot study with valinomycin", the authors propose a method for high-throughput genotoxicity testing based on γH2AX detection using immunofluorescence microscopy and automated bioimage analysis in open-source software, aplying several simplifications to their previously published work.

The authors manage to prove that overall fluorescence intensity of γH2AX obtained from bioimage analysis appears to be a promising alternative to flow cytometry in genotoxicity testing in vitro. Also, the paper is clearly organized and well written.

I believe that it can be accepted for publication in MDPI Toxins, after some very minor changes:

-          - page 10 – lines 292-293: replace « The authors measured median FL1…. » with « We measured median FL1… » ;

-          - I was wondering if the References part could be more up to date, since approx. 50% of the references cited are more than 10 years old ?

Author Response

Dear sir or madame,

we have finished the revisions upon requests:

1) page 10 – lines 292-293: replace « The authors measured median FL1…. » with « We measured median FL

Answer:

We refer here to the work of Smart et al.(2011), we have changed the sentence to avoid the misunderstanding.

2) I was wondering if the References part could be more up to date, since approx. 50% of the references cited are more than 10 years old ?

Answer:

The references were updated, however we are trying to cite the primary sources, thus older references are cited.

We added e.g. Rahmanian et al (2021), Zhang et al (2023)